# Metagenomics-Based Proficiency Test of Smoked Salmon Spiked with a Mock Community

**DOI:** 10.3390/microorganisms8121861

**Published:** 2020-11-25

**Authors:** Claudia Sala, Hanne Mordhorst, Josephine Grützke, Annika Brinkmann, Thomas N. Petersen, Casper Poulsen, Paul D. Cotter, Fiona Crispie, Richard J. Ellis, Gastone Castellani, Clara Amid, Mikhayil Hakhverdyan, Soizick Le Guyader, Gerardo Manfreda, Joël Mossong, Andreas Nitsche, Catherine Ragimbeau, Julien Schaeffer, Joergen Schlundt, Moon Y. F. Tay, Frank M. Aarestrup, Rene S. Hendriksen, Sünje Johanna Pamp, Alessandra De Cesare

**Affiliations:** 1Department of Physics and Astronomy, University of Bologna, 40127 Bologna, Italy; claudia.sala3@unibo.it; 2Research Group for Genomic Epidemiology, National Food Institute, Technical University of Denmark, Kemitorvet, DK-2800 Kgs, 2800 Lyngby, Denmark; hamr@food.dtu.dk (H.M.); tnpe@food.dtu.dk (T.N.P.); casper.sahl.poulsen@sund.ku.dk (C.P.); fmaa@food.dtu.dk (F.M.A.); rshe@food.dtu.dk (R.S.H.); sjpa@food.dtu.dk (S.J.P.); 3German Federal Institute for Risk Assessment, Department of Biological Safety, 12277 Berlin, Germany; Josephine.gruetzke@bfr.bund.de; 4Highly Pathogenic Viruses, ZBS 1, Centre for Biological Threats and Special Pathogens, Robert Koch Institute, 13353 Berlin, Germany; BrinkmannA@rki.de (A.B.); NitscheA@rki.de (A.N.); 5Teagasc Food Research Centre, Moorepark, APC Microbiome Ireland and Vistamilk, T12 YN60 Co. Cork, Ireland; paul.cotter@teagasc.ie (P.D.C.); fiona.crispie@teagasc.ie (F.C.); 6Surveillance and Laboratory Services Department, Animal and Plant Health Agency, APHA Weybridge, Addlestone, Surrey, KT15 3NB, UK; Richard.Ellis@apha.gov.uk; 7Department of Experimental, Diagnostic and Specialty Medicine, University of Bologna, 40127 Bologna, Italy; gastone.castellani@unibo.it; 8European Molecular Biology Laboratory, European Bioinformatics Institute, Wellcome Genome Campus, Hinxton, Cambridge CB10 1SD, UK; amid@ebi.ac.uk; 9National Veterinary Institute, Ulls väg 2B, 75189 Uppsala, Sweden; mikhayil.hakhverdyan@sva.se; 10Laboratoire de Microbiologie, CEDEX 03, 44311 Nantes, France; soizick.le.guyader@ifremer.fr (S.L.G.); julien.schaeffer@ifremer.fr (J.S.); 11Department of Agricultural and Food Sciences, University of Bologna, 40064 Ozzano dell’Emilia, Italy; gerardo.manfreda@unibo.it; 12Epidemiology and Microbial Genomics, Laboratoire National de Santé, L-3555 Dudelange, Luxembourg; joel.mossong@lns.etat.lu (J.M.); catherine.ragimbeau@lns.etat.lu (C.R.); 13Nanyang Technological University Food Technology Centre (NAFTEC), Nanyang Technological University (NTU), 62 Nanyang Dr, Singapore 637459, Singapore; jschlundt@ntu.edu.sg (J.S.); moon.tay@ntu.edu.sg (M.Y.F.T.); 14Department of Veterinary Medical Sciences, University of Bologna, Via Tolara di Sopra 50, 40064 Ozzano dell’Emilia, Italy

**Keywords:** shotgun metagenomics, smoked salmon, proficiency test, experimentally spiked samples, wet-lab protocols

## Abstract

An inter-laboratory proficiency test was organized to assess the ability of participants to perform shotgun metagenomic sequencing of cold smoked salmon, experimentally spiked with a mock community composed of six bacteria, one parasite, one yeast, one DNA, and two RNA viruses. Each participant applied its in-house wet-lab workflow(s) to obtain the metagenomic dataset(s), which were then collected and analyzed using MG-RAST. A total of 27 datasets were analyzed. Sample pre-processing, DNA extraction protocol, library preparation kit, and sequencing platform, influenced the abundance of specific microorganisms of the mock community. Our results highlight that despite differences in wet-lab protocols, the reads corresponding to the mock community members spiked in the cold smoked salmon, were both detected and quantified in terms of relative abundance, in the metagenomic datasets, proving the suitability of shotgun metagenomic sequencing as a genomic tool to detect microorganisms belonging to different domains in the same food matrix. The implementation of standardized wet-lab protocols would highly facilitate the comparability of shotgun metagenomic sequencing dataset across laboratories and sectors. Moreover, there is a need for clearly defining a sequencing reads threshold, to consider pathogens as detected or undetected in a food sample.

## 1. Introduction

Foodborne illnesses associated with pathogenic microorganisms are a global public health and economic challenge. In 2018, 26 European Member States reported 5146 food-borne and waterborne outbreaks with 48,365 human cases and 21.2% increase in the overall number of outbreak-related deaths, in comparison to 2017 [1]. The outbreaks with known causative agents were caused by bacteria (57.0%), followed by bacterial toxins (24.2%), viruses (13.5%), other causative agents (4.3%), and parasites (1.0%) [1]. The techniques used for the detection and characterization of foodborne pathogens in food products evolved tremendously over the past several decades, but they generally focused on the detection of a single pathogen or just a few pathogens at a time [2]. This analytical restriction hampers the mapping of shifting microbial communities, which potentially affect the persistence of foodborne pathogens in the food production chain [2,3], and can also result in pathogens being overlooked by virtue of being novel or not being traditionally associated with a particular environment.

Shotgun metagenomics provides the potential for detection, identification, and characterization of pathogens in food [4,5], in the food chain environment [2], as well as in animals and humans [6,7]. In addition to taxonomic assignment, shotgun metagenomics also provides functional insights, through the detection of genetic markers, such as genes associated with antimicrobial resistance and virulence-related properties [5]. However, at present, deconvoluting the metagenomic data to definitively associate those markers with the same genome is still challenging [5,8].

Shotgun metagenomic sequencing includes a wet-lab part, followed by sequence data analysis. Concerning the wet-lab protocols, comparative studies were performed to assess the implementation of different strategies for sample handling [7,9], nucleic acid extraction [10], library preparation [11], and sequencing [12]. Furthermore, specific assessments were done on working with contaminants [13], host DNA [14], and low biomass specimens [6], defining ideal read depths for particular biological specimens or food products, as well as defining the sequence number thresholds to confidently assign a pathogenic etiology [15]. Nowadays, Giga bases of high-quality sequence data can be easily generated at a comparatively low cost. Thus, performing high-throughput shotgun sequencing can result in a large and complex dataset from which taxonomic composition and functional capacity of the entire ecosystem under study can be determined [16].

Various pipelines for the pre-processing, assembly [17], clustering, and analyses are available for metagenomic bioinformatics, such as IMP [18], MetAMOS [19], MG-Mapper [20], MG-RAST [21], MOCAT2 [22], OneCodex [23], and RIEMS [24]. However, in some cases, the use of those pipelines requires access to high performance computing facilities, as well as laboratory personnel that is cross-trained in bioinformatics and biology, to generate the data and interpret the results obtained [8].

To access the suitability of shotgun metagenomics to detect and possibly quantify microorganisms belonging to different domain in a food sample, a proficiency test (PT) was carried out as part of the COMPARE project (www.compare-europe.eu), involving 11 participants from the EU and Singapore. All participants received the same food sample spiked with a mock community composed of six bacteria, three viruses, one parasite, and one yeast. Each participant processed the sample using in-house wet-lab protocol(s), up to the production of the metagenomic dataset(s). Metagenomic datasets were both submitted to a COMPARE data-hub [25] and picked up or sent directly to the participant performing the data analysis, using the MG-RAST metagenomics analysis server. The relative abundance of the microorganisms of the mock community expected in each dataset was calculated, based on the number of cells/viral genomes spiked in the food sample. Moreover, the detection of each microorganism was evaluated, considering the presence of at least 1, 5, or 10 corresponding reads enumerated in MG-RAST. The DNA metagenomic datasets were ranked according to the distance between the relative abundance of the microorganisms of the mock community in each dataset, and the expected values. The best-performing wet-lab protocols generating the metagenomic datasets closer to the expected values were then discussed.

## 2. Materials and Methods

### 2.1. Samples

An aliquot of cold smoked salmon was cut in small pieces (approximately 1–2 mm in width/length/depth), using a sterile scalpel and sterile petri dish. An amount of 0.2 g was transferred to individual sterile Nunc screw cap tubes. Subsequently, each tube was kept on ice and spiked with 50 µL of a mock microbial community, consisting of six bacteria, one parasite, one fungus, and 10 µL of heat-inactivated viruses (one DNA and two RNA viruses) (Table 1).

The tubes were kept on ice during preparation. Cell counts of the bacteria and the fungus were determined using a Petroff counting chamber, under a light microscope, counting cells in two diagonal corners on two separately prepared slides. The parasite oocysts were obtained from Waterborne Inc. in PBS, containing 1.2 × 108 cells (as determined by Fluorescence Activated Cell Sorting, by the provider). After spiking, each tube was vortex-mixed and placed at −80 °C, before shipping to each PT participant with a freezer pack kept at −80 °C, until packaging. The frozen samples were supplemented by another 50 µL of the virus mix, without additional mixing.

### 2.2. Laboratory Workflows of Participants

The wet-lab protocols used by the participants differed from each other. These are summarized in Table 2 and are detailed in the Appendix A.

A total of 27 metagenomics datasets obtained by shotgun metagenomic sequencing of spiked salmon were submitted as part of the PT. Among the 27 samples tested, 7 were not pre-processed and directly submitted to nucleic acid extraction, while 9 were pre-processed by using a bead beating protocol with TissueLyser, with modifications by each participant. Moreover, for M33 and M34, two milliliter of glycine buffer 0.05 M (pH 9) were added to the spiked salmon and homogenized in a potter grinding tube. Then, the pH was decreased to 3 by adding HCL. An equal volume of chloroform-butanol (v/v) was added and mixed by vortexing. After centrifugation, the supernatant was treated with Cat-Floc T (Calgon, Ellwood City, PA, USA) and precipitated with polyethylene glycol 6000 (PEG 6000) (Sigma, St. Quentin, France), for 1 h at 4 °C, and centrifuged for 20 min at 11,000× *g* at 4 °C. The pellet was resuspended in 2 mL of glycine buffer and filtrated using a cascade of 5, 1.2, and 0.45 mm filter pores (Minisart NML 17594, NML17593, PES16533, and PES16532). The recovered filtrates were incubated for 1 h at 37 °C, with 2000 Units of OmniCleave EndonucleaseTM (Lucigen corporation) and 100 mL of MgCl2 (100 mM). For M30, M31, and M32, the samples were centrifuged at 5000 rpm for 20 min at 4 °C, and the pellet was used for nucleic acid extraction. For M08, the sample was homogenized in liquid nitrogen using a ceramic mortar and pestle, followed by the gradual addition of 2.5 mL TE buffer. The homogenate was then centrifuged at 4000× *g* for 10 min and the supernatant was filtered on a 0.45-µm disc HPF Millex syringe filter (Millipore, Cork, Ireland), followed by the nuclease treatment. Subsequently, 200 µL of the sample were used for DNA extraction. For M11 and M13, the samples were homogenized in liquid nitrogen, using a ceramic mortar and pestle, followed by the gradual addition of 2.5 mL TE buffer. The homogenate was then centrifuged at 4000× *g* for 10 min, and the supernatant was subjected to three cycles of the freeze–thaw method (M11: dry ice/100 °C; M13: dry ice/37 °C) and submitted to DNA extraction. For M12, the sample was homogenized in liquid nitrogen, using a ceramic mortar and pestle, followed by the gradual addition of 2.5 mL TE buffer. The homogenate was then centrifuged at 4000× *g* for 10 min, and the supernatant was filtered on a 0.45-µm disc HPF Millex syringe filter (Millipore, Cork, Ireland), followed by nuclease treatment. Subsequently, 200 µL of the sample was transferred to a new tube and mixed with 600 µL of TRIzol Reagent (Invitrogen, Carlsbad, CA, USA), 160 µL of chloroform, followed by RNA extraction. Finally, for M27 and M28, the samples were submitted to a disintegration step, using the Covaris cryoPREP CP02 [7].

For 18 samples, the target nucleic acid was DNA (Table 2). In 2 samples, the DNA was extracted using the QIAamp Fast DNA Stool Mini Kit (Qiagen, Hilden Germany) and in 3, using the DNeasy PowerSoil Microbial Kit (Qiagen). Moreover, in 5 samples, the DNA was extracted using the DNeasy PowerFood Microbial Kit (Qiagen) and in 8 samples using the QIAamp DNA Mini Kit (Qiagen) or the QIAamp UCP Pathogen Mini Kit (Qiagen). In M08, M11, and M13, the QIAamp DNA Mini Kit Qiagen was associated to Tag labeling and random amplification (SISPA) [26].

In 9 samples, the target nucleic acid was RNA (Table 2). In one sample, the RNA was extracted using the Direct-zol RNA Kit (Zimo Research) and in 2, the NucliSENS^®^ miniMAG^®^ (BioMérieux, Marcy l’Etoile, France). Moreover, in 2 samples, the RNA was extracted using the QIAamp Viral RNA Mini Kit (Qiagen) and in 4, the RNeasy Mini Kit (Qiagen). For M12, the use of the kit was associated to Tag labeling and Sequence-Independent, Single-Primer-Amplification (SISPA). In 5 samples, the extracted RNA was reverse transcibedusing the cDNA Synthesis System Kit (Roche, Basel, Switzerland), while in 4, the SuperScript IV Reverse Transcriptase (Invitrogen Thermo Fischer Scientific (Walthman, MA, USA) was used.

For the library preparation, 19 samples were processed using the Nextera™ XT DNA Library Prep kit (Illumina) and in one, the Nextera™ DNA Flex Library Preparation kit (Illumina, San Diego, CA, USA) was used. The libraries for the remaining 7 samples were prepared using the TruSeq^®^ DNA Library Prep Kit Illumina (n = 3), the NEBNext^®^ Ultra™ II DNA Library Prep Kit for Illumina^®^ (New England BioLabs, Ipswich, MA, USA) (n = 2), and the GeneRead DNA Library kit Qiagen (n = 2) (Table 2).

All samples were sequenced in paired-end, except two (Table 2), and the read lengths were achieved by sequencing ranged between 120 and 300 bp (Table 2). A total of 16 samples were sequenced on the NextSeq500 (Illumina), while the others were sequenced on the HiSeq2500 (Illumina) (n = 1), MiSeq (Illumina) (n = 7), MiniSeq (Illumina) (n = 1), and Ion S5XL System (Thermo Fischer Scientifics) (n = 2) (Table 2). The sequencing output associated with each dataset ranged between 0.86 and 12.2 Gbp.

### 2.3. Data Sharing

All metagenomic datasets included in this PT are publicly available on https://www.mg-rast.org/mgmain.html?mgpage=project&project=mgp86519 in the MG-RAST server, under the project labelled as Food metagenomic ring trial 2018. The metadata associated with the metagenomic datasets are those detailed in Table 2.

### 2.4. Bioinformatics and Statistical Analysis

The workflows including DNA as the target nucleic acid were exploited in the modelling and bio-statistical analysis described below, while RNA-based metagenomic datasets were only used to evaluate the detection performances, since the lack of information on RNA copy numbers prevented the possibility of computing quantitative comparisons. For the bio-statistical analysis, similar wet-lab steps were categorized together (Table 2).

Filtering, trimming, and taxonomic classification of raw reads were performed using MG-RAST (https://www.mg-rast.org) [27] and the RefSeq reference database [28]. The statistical analysis was performed in R, v3.5.1, using the libraries phyloseq v1.26.1 [29] and DESeq2 v1.22.1 [30].

Before proceeding with the statistical analysis, the read counts were normalized using DESeq2 [30], taking into account the compositional nature of the data. In brief, starting from the whole read counts table, DESeq2 estimates the size factor of each sample, as the median of the ration of the observed counts to those of a reference sample, obtained by taking the geometric mean across samples. Then, the normalized counts are obtained by dividing the original counts by the estimated size factors. Finally, DESeq2 computes the dispersion estimates of each gene using an Empirical Bayes approach. Such estimates are then exploited in the negative binomial generalized linear models used in the differential analysis [30,31].

The expected proportions of the microorganisms in the mock community and those detected in the metagenomics datasets submitted as part of the PT, were compared using the following approach. The expected relative abundances of the mock community were calculated, based on the cell counts of each spiked microorganism (Table 1). Such concentrations were first multiplied by the genome length to simulate the fact that in a sequencing experiment, longer genomes would be represented by a higher number of reads. Then, relative abundances were obtained by normalizing the resulting values to sum of one. The empirical relative abundances obtained in the 27 DNA metagenomic datasets were computed, based on the number of reads that mapped on the mock community microorganisms. Specifically, for each sample, the normalized abundances were first obtained with DESeq2 and the relative abundances of the mock community microorganisms were computed by normalizing the sum to one such abundances.

To visually compare the expected and the experimentally obtained compositions of the mock community in the DNA metagenomic datasets, we plotted the relative abundances in the form of bar plots. Differences in the mock community composition among samples and between each sample and the expected composition, were also evaluated by computing the Bray-Curtis beta-diversity displayed using the heat map. Samples were then ranked based on their distance (i.e., Bray-Curtis beta-diversity) from the expected composition of the mock community.

The statistical comparison of the species abundances obtained with different experimental parameters was performed with DESeq2. Specifically, starting from the normalized read counts, DESeq2 was used to compute the negative binomial generalized linear models for each taxon and, hence, to test the impact of the variables of interest.

We evaluated a multiple regression model in which we considered all experimental parameters in order to detect the effect of each variable, after adjustment for all others. To select the model variables, we first estimated the relationship between the variables using Fisher’s exact test for categorical data and the slope of a linear regression model for numerical variables. The only pair of variables for which the *p*-value was <0.0001 was read length-sequencing platform (*p*-value = 4.47 × 10^−5^). In all other cases, the *p*-value was ≥0.0001 and both variables were hence included in the model. All in all, the tested variables were pre-processing protocol, DNA extraction kit, library preparation strategy, sequencing depth, and sequencing platform. The advantage of using a multiple regression model that considers all predictors (experimental variables) at the same time is that, when evaluating the effect of one predictor, it adjusts for the effect of all others. The resulting estimates of this model, sometimes called ‘partial effects’, hence, indicate the effect that each experimental variable has on the abundances of a certain species, when all other variables are fixed. Statistical significance was assessed using the Likelihood Ratio Test for multiple comparisons and the Wald’s test for pairwise comparisons. Moreover, the impact of each workflow on the detected mock community microorganisms’ abundances was evaluated by performing a pairwise two-samples *t*-test between the DESeq2 normalized read counts, obtained with each pair of workflows. For bacteria, the parasite and the yeast, such a *t*-test was computed by considering the mean value and the variance of the abundances detected in the workflows’ samples. Since only one sample was tested by applying WF12, in this case, the sample mean of each species was set to the value of its abundance in that sample and the sample variance was estimated as the average of the variances observed for that species, in all other workflows. Finally, the sample size was set to the average number of samples present in the other workflows. In this approximated test, we assume that the variance of each microorganism’s abundance among the samples within a workflow is the same for all workflows with only small deviations (i.e., variances of the microorganisms’ abundances are comparable between different workflows), as shown in Appendix A. In all tests, the p-values were adjusted for multiple testing using the Benjamini–Hochberg procedure [32] and a significance level of 0.05 was used to identify the statistically significant differences.

## 3. Results

### 3.1. Relative Abundance of the Reads Assigned to the Taxonomic Domains and the Microorganisms of the Mock Community

The reads belonging to different taxonomic domains were quantified in the 27 metagenomics datasets, submitted as part of the PT (Table 3).

The percentage of reads assigned to Eukaryota, including the two microorganisms belonging to the mock community (i.e., *C. parvum* and *S. cerevisiae*) ranged between 2.957% (M34) and 80.567% (M11). The reads assigned to the domain bacteria ranged between 18.823% (M11) and 95.202% (M36), and those assigned to viruses ranged between 0.031% (M19) to 38.344% (M33) (Table 3). These results showed that the reads assigned to each domain largely differed among the metagenomic datasets and this affected the reads assigned to the microorganisms of the mock community. While it is difficult to explain each of these differences, in the three datasets in which the percentage of reads assigned to the viruses was ≤0.1 (i.e., M19, M20, and M28), RNA was extracted using the RNeasy mini kit.

The quantification of the mock community microorganisms, in terms of relative abundance, differed among the DNA metagenomic datasets (Appendix A). As detailed in Materials and Methods, the expected relative abundance of each microorganism was calculated, based on the number of cells and virus genome copies experimentally spiked in the salmon, and the microorganisms’ genome size. The relative abundances obtained from the 18 metagenomic datasets were compared to each other and to the expected one (Figure 1), and the pairwise Bray–Curtis dissimilarity among samples and with the expected composition was computed (Figure 2).

Appendix A shows the reads enumerated for each microorganism of the mock community using MG-RAST. The read number threshold to consider a microorganism as detected or undetected in a metagenomic dataset is currently unknown. Nevertheless, Appendix A summarizes the percentage of metagenomic datasets obtained from DNA and RNA in which each microorganism of the mock community was detected by assessing the enumeration in MG-RAST of at least 1, 5, or 10 corresponding reads, as the detection threshold. All spiked bacteria, the yeast, and the parasite, were always detected in the metagenomic datasets from DNA, while the percentage of metagenomic datasets in which *C. parvum* was detected decreased from 100 to 67% in the metagenomic dataset from RNA, increasing the detection threshold from one to 10 reads. The same result applied to *F. nucleatum* and *P. freudenreichii* detected in 100% of the metagenomic datasets from RNA, by considering one read as the detection threshold, while in 78% 10 reads were considered as the detection threshold (Appendix A). In relation to viruses, the DNA virus was detected in 83% of the metagenomic datasets from DNA but also in 56 to 33% of those from RNA. On the contrary, the RNA viruses were hardly detected in RNA metagenomic datasets, when even 1 read was considered as the detection threshold, and it did not show up at all in the DNA metagenomic datasets (Appendix A).

### 3.2. Ranking of the Metagenomic Datasets Based on Their Dissimilarity to the Expected Composition and Assessment of the Impact of Each Variable of the Workflow on the Abundance of the Mock Community Members

Considering the relative abundances of all microorganisms of the mock community, the DNA metagenomic datasets were ranked according to their Bray–Curtis distance from the expected values. The metagenomic dataset ranking closest to the expected value was M36 (Bray–Curtis distance 0.199) followed by M38 (Table 4), and this ranking did not change when considering the bacteria only (i.e., M36 Bray–Curtis distance 0.152) (Appendix A).

Among the variables investigated in the workflows, those impacting on the abundance of one or more microorganisms spiked in the salmon were the pre-processing protocol, the DNA extraction protocol, the library preparation strategy, and the sequencing platform.

The pre-processing protocol significantly affected the detected abundances of *C. parvum*, *E. coli*, *F. nucleatum*, *P. freudenreichii*, *S. enterica,* and *S. aureus* (Appendix A). The normalized mean abundances of the parasite and *F. nucleatum* were significantly higher in the metagenomic datasets obtained without pre-processing, in comparison to where BBTL was applied, while the application of a bead beating protocol provided the best results in terms of *E. coli*, *P. freudenreichii*, *S. enterica,* and *S. aureus* abundances detected (Appendix A). Moreover, for *F. nucleatum,* the application of any pre-processing protocol worked better than all the other tested pre-processing protocols (Appendix A).

The abundances of *C. parvum*, *B. fragilis*, *E. coli*, *P. freudenreichii*, *S. enterica,* and *S. aureus* were significantly higher in the metagenomic datasets where the DNA was extracted by PowerFood, in comparison to the QIAamp DNA Mini Kit, with or without SISPA or the QIAamp UCP Pathogen Mini Kit, while for *F. nucleatum*, the latter performed better (Appendix A). On the other hand, for both *F. nucleatum* and *S. aureus,* the QIAamp DNA Mini Kit, with or without SISPA or the QIAamp UCP Pathogen Mini Kit, performed better than QIAamp Fast DNA Stool Mini Kit and the DNeasy PowerSoil Microbial Kit, while for *B. fragilis*, *E. coli*, *P. freudenreichii,* and *S. enterica,* it was the opposite (Appendix A).

The abundances of *E. coli*, *F. nucleatum*, *P. freudenreichii*, *S. enterica,* and *S. aureus* were significantly affected by the library preparation protocol. Nextera™ XT DNA Library Prep kit and Nextera™ DNA Flex Library Preparation kit worked better for *E. coli*, while TruSeq^®^ DNA Library Prep Kit, NEBNext^®^ Ultra™ II DNA Library Prep Kit for Illumina^®^ and the GeneRead DNA Library kit, resulted in better detected abundances for *F. nucleatum*, *P. freudenreichii*, *S. enterica,* and *S. aureus* (Appendix A).

The yeast abundance was not affected by the sequencing platform, while the abundances of all other microorganisms was significantly higher in the metagenomic datasets obtained by sequencing on a NextSeq500 or a HiSeq 2500, in comparison to other sequencing platforms (i.e., MiSeq, MiniSeq, Ion S5XL System) (Appendix A).

### 3.3. Assessment of the Impact of Each Workflow on the Abundance of the Mock Community Members

The results for each workflow (WF), starting with DNA as a target nucleic acid were compared by performing a pairwise approximated t-test, in which we tested if the mock community microorganism’s abundances detected by each pair of workflows differed.

The average abundance of the parasite was significantly higher for the samples processed using the WFs 2 and 6, in comparison to WF11 (Appendix A). In the WF2, the salmon is transferred to a Power Bead Tube inserted into the preheated (55 °C) adapter of TissueLyser II and bead-beating was applied, before centrifugation at full speed for 1 min. Subsequently, the supernatant was transferred to a new 2 mL tube before DNA extraction, using the MoBio PowerFood Microbial DNA Isolation kit. In the WF6, the salmon was centrifuged at 5000 rpm for 20 min at 4 °C, and then processed as in WF2. The main differences between WF11 in comparison to WFs 2 and 6 was that sequencing was performed on the longer reads (i.e., 300 vs. 150 bp) and, on average, at a lower depth (2 vs. >7 Gbp). Moreover, as detailed in the Appendix A, the sample handling protocol in WF11 was very long and might have negatively impacted the DNA recovery. The significantly low performances of WF11, in comparison to all other WFs, was also quantified for *F. nucleatum*. The average abundance of the yeast was significantly higher for the samples processed using WF4, including a bead beating protocol with TissueLyser than for WF6, while for *B. fragilis* it was the opposite (Appendix A).

## 4. Discussion

Shotgun metagenomics is a culture-independent methodology with potential to contribute to food-borne outbreaks detection and risk assessment of food-borne pathogens [33]. It consists of a wet-lab part and a dry-lab part. The wet-lab part includes the sample collection for the generation of the raw sequencing dataset. The dry-lab part includes the bioinformatic analysis, resulting in taxonomic and functional gene assignments, using reference or de novo strategies, and the biostatistics analysis, translating the bioinformatic results in biologically meaningful observations. Many publications are available on different bioinformatic strategies to trim, assemble, and assign reads and contigs, to known or unknown taxonomic and functional entities [34]. However, there are few fully accessible and transparent bioinformatic workflows and many of the online available bioinformatic tools for non-experts are not constantly updated in the reference databases. Biostatistics strategies are complex and thus they must be selected with great care, when modelling and analyzing the metagenomic datasets. Therefore, the pipelines fitting specific scenarios and case studies should be implemented and shared at an international level, to improve harmonization in metagenomic dataset analysis and interpretation.

This PT was organized with the aim to compare the suitability of different metagenomic wet-lab protocols, for both detecting and quantifying the relative abundance of microorganisms belonging to the different domains that were experimentally spiked in cold smoked salmon. Although a drawback of this study was that the large number of variables included in the wet-lab workflows applied by the participants resulted in the need to categorize similar but not identical wet-lab steps, nonetheless, this PT was the first exercise on a real food matrix, while other studies addressed artificial datasets. Moreover, to make the results of this PT as comprehensive as possible, the pipeline used for the bioinformatic analysis (i.e., MG-RAST) is a freely available and user-friendly web resource, while more powerful bioinformatic tools could have been used. The outputs of this PT should contribute to speed up the application of shotgun metagenomic in food microbiome studies and in food safety risk assessment, but for both, a quality assurance approach consistent with regulatory standards is mandatory. At present, individual research institutes set the internal reference standards, but the lack of universal reference organisms and genomic material makes it difficult to compare assay performances between different laboratories [35]. Hence, it is unlikely that shotgun metagenomic sequencing will replace the culture methods for foodborne pathogen investigation in the near future, although its great potential was demonstrated [5,36,37].

The main methodological constraints to be overcome are the lack of harmonized and validated methods, the low sensitivity in detecting certain taxa in the sample, or the fact that the results obtained strongly depend on the choice of wet-lab workflow and the bioinformatics pipelines, as shown by our results. The scientific community must agree on the basic requirements to consider a dataset meaningful for both bioinformatic and biostatistics analysis. If such requirements would be clearly defined, then different workflows could be applied for the wet-lab part, as far as those requirements are achieved, exploring further protocols to reduce shotgun metagenomic sequencing costs and improving efficiency [38].

In this PT, the RNA viruses were hardly detected or not detected at all in the metagenomic datasets. There are different possible reasons for the observed failure to detect viruses. This is likely caused by the low relative spiking input of the viruses, in comparison to the final overall nucleic acid content, which was intended to resemble a situation of exogenous viral contamination. However, a pilot sequencing of a salmon aliquot after sample preparation with one-sixth (10 µL) of the virus mix, resulted in a very low recovery for the viruses. Therefore, before the samples were sent out to the participants, another 50 µL (five-sixth of the final amount of virus mix) was added to the frozen sample. Moreover, it should be mentioned that the necessary heat-inactivation of the virus mix could cause the virus particles to be destroyed, leading to a release of unprotected viral nucleic acids. Ultimately, this could have caused a loss of viral nucleic acids, before sample processing by the participants, due to exposure to the nucleases present in the salmon matrix. To solve the issue of low virus concentration, which is common to food and human samples, it was suggested that an RNA extraction be carried out and then to combine targeted with untargeted sequencing, spiking the variably sized panels (100–10,000) of short primers into the reaction mixtures, at the reverse transcription step [39]. Following this approach, Thézé, J. et al., 2018 increased the number of Zika virus reads by more than tenfold, without substantially decreasing sensitivity for other pathogens in the metagenome. An enrichment during the library preparation, also allowed virus identification in foods like oysters [40].

Considering the relative abundances of all microorganisms of the mock community, the DNA metagenomic datasets were ranked according to their Bray–Curtis distance from the expected values. Although it was shown that MG-RAST has good sensitivity and precision when assigning reads to the species of a mock community [41], we acknowledge that MG-RAST suggests against the usage of the species taxonomic level, and that the bioinformatic processing of data might have affected the quantification of the species of the mock community, especially in the presence of possible contamination microorganisms that are close relatives of the mock community species. The DNA metagenomic ranking at the highest position (i.e., Bray–Cutis distance <0.20) were M36, followed by M38 characterized by 95 and 78% of the total reads assigned to the domain bacteria (Table 3), which were the main representatives of the mock community. In M36, the DNA was obtained using the QIAamp DNA Mini Kit (Qiagen) supplemented with proteinase K, the TruSeq Nano DNA Library Preparation protocol, and sequenced on an Illumina HiSeq2500 sequencer in rapid mode, at a read-length of 250 bp paired-end to a coverage of 8.88 Gbp. With regards to the library preparation strategies, Grützke et al., 2019 [42] showed that genomes from bacterial species within a mock community were better detected using the TruSeq Nano, the Nextera Flex, and the TruPLEX kits over the Nextera XT kit. This result was explained with a shift in the GC density of fragments, generated with the Nextera XT kit to higher GC-contents, while this was balanced for the other three library kits, and reflected the expected distribution from the collection of mock-community reference genomes [41]. This specific issue was not investigated in this PT, although we also discovered a significant positive impact of the TruSeq Nano DNA Library Prep kit, on the quantification of *S. aureus* (Appendix A), which feature a genome with a low GC-content.

Among the variables investigated in the workflows, those impacting on the abundance of one or more microorganisms spiked in the salmon were the pre-processing protocol, the DNA extraction protocol, the library preparation strategy, and the sequencing platform. These results were certainly affected by the multi regressing modelling approach, quantifying the effect of each experimental variable (e.g., pre-processing) on the abundances of a certain species, when all other variables were fixed, which did not reflect what happened in real life, where each step of the wet-lab protocol impacts on the following. However, since each participant applied in house wet-lab protocols without restriction, the resulting variables in the metagenomic datasets were many, and the multi regressing modeling approach allowed us to collect meaningful results, species by species. This limitation might explain why sequencing coverage did not turn out to be a key variable affecting the relative abundance of the spiked organisms, although the six metagenomic datasets closest to the expected composition of the mock community were all sequenced at ≥8 Gbp. Clear indications on the sequencing coverage threshold to be achieved in metagenomic investigations are lacking. However, Ni et al., 2013 [43] stated that to detect bacteria species with a relative abundance of more than 1%, a 20× coverage should be obtained, corresponding to a sequencing depth of 7.15 Gbp.

The investments in research on shotgun metagenomics are justified by the fact that the results achieved in both human and food sectors translate in the identification of potential pathogens in the ecosystems where those pathogens are in real life, and as a matter of fact, the interaction between pathogens and their ecosystems affects both pathogen survival and multiplication ability [2,44]. If sample preparation is designed to be as non-specific as possible to capture all nucleic acids regardless of their source, shotgun metagenomics is applicable simultaneously for viruses, bacteria, and parasites, as was found in this study for several workflows and according to previous reports [7]. In this framework, it is important to highlight that in the annual EFSA-ECDC report on foodborne and waterborne outbreaks occurring in the EU, there is always a high percentage of outbreaks for which the causative agent is described as ‘unknown’ or ‘unspecified’. In the last available report, referring to the outbreaks which have occurred in 2018 in 28 Member States (MS) and 8 non-MS, such a percentage was as high as 23.8% [1] and some of these unknown were likely to be uncultivable or difficult to culture microorganisms, which could possibly be detected using shotgun sequencing. However, for diagnostic metagenomics to become truly useful, the method must provide robust and reproducible outputs [31].

In conclusion, our results showed that there are huge differences in the workflows applied in the wet-lab part of shotgun metagenomic sequencing at the international level, while there is a need for harmonized and validated protocols. Despite the differences between wet-lab protocols, all microorganisms of the mock community, belonging to the different domains (i.e., bacteria, parasite, yeast, DNA, and RNA viruses) were successfully detected in the metagenomic datasets obtained from spiked samples, suggesting that different wet-lab protocols could be successfully applied to reach the same result. Nonetheless, since the percentages of metagenomic datasets in which the mock community members were detected changed according to the number of reads selected as the detection cut-off level, this parameter should be clearly defined to use shotgun metagenomic sequencing in food microbiome studies, and in food safety risk assessment. There are many valuable papers on the application of shotgun metagenomics, but the lack of transparent information on the technical details of both the wet-lab and bioinformatic procedures are delaying the full implementation of this powerful sequencing approach.

## Figures and Tables

**Figure 1 microorganisms-08-01861-f001:**
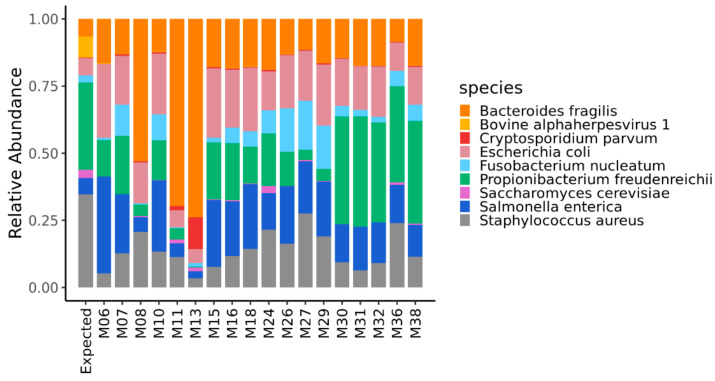
Relative abundance of the microorganisms of the mock community quantified in the DNA metagenomic datasets obtained from spiked salmon. The first bar in the figure refers to the expected relative abundance for the microorganisms experimentally spiked in the salmon.

**Figure 2 microorganisms-08-01861-f002:**
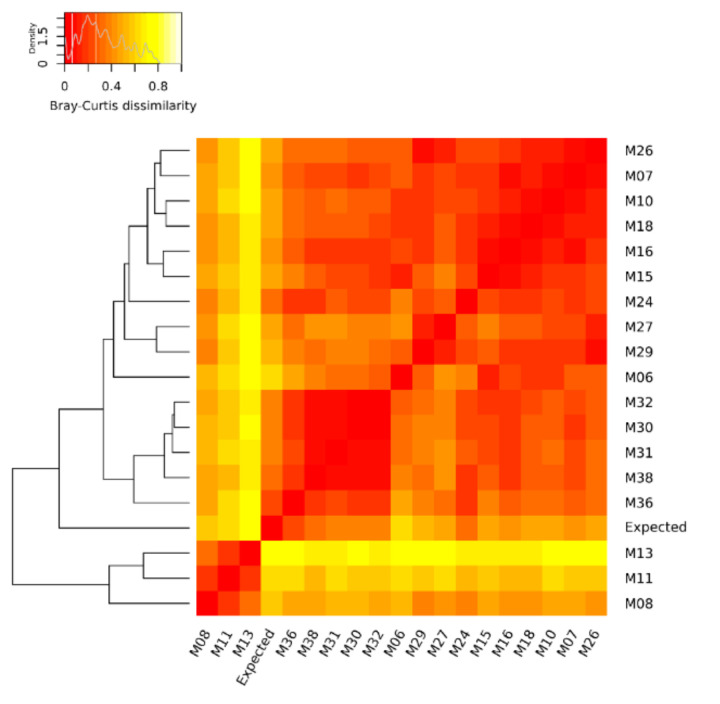
Heat map of Bray–Curtis dissimilarity and average linkage clustering dendrogram showing the similarity between the benchmark (expected) and the DNA metagenomic datasets obtained from spiked salmon, considering the microorganisms of the mock community. In the heat map, when a cell is colored in red, it indicates that the two associated samples (in the corresponding row and column) have an identical composition, while a white cell refers to two samples with the most different composition.

**Table 1 microorganisms-08-01861-t001:** Composition of the mock community used to spike the samples of cold smoked salmon and concentration of each microorganism. Each concentration corresponded to an expected relative abundance calculated by the number of cells of each spiked microorganism multiplied for the genome size, as indicated in the GeneBank (https://www.ncbi.nlm.nih.gov/assembly), taking the value available or the mean * of the different values available for the same strain. The relative abundances were obtained by normalizing the resulting values to a sum of one.

Taxon (Genome Size)	Number per Subsample (Cells/Virus Genome Copies)	Expected Relative Abundance	Feature
Bacteria			
*Bacteroides fragilis* NCTC 9343/DSM 2151 (5,241,700 bp)	5 × 10^7^	0.065	Gram −
*Escherichia coli* ATCC 25922 (5,166,282 * bp)	5 × 10^7^	0.064	Gram −
*Fusobacterium nucleatum* subsp. *nucleatum* ATCC 25586/DSM 15,643 (2,177,300 * bp)	5 × 10^7^	0.027	Gram −
*Propionibacterium freudenreichii* subsp. *Freudenreichii* DSM 20271 (2,649,166 bp)	5 × 10^8^	0.331	Gram +
*Salmonella enterica* subsp. *enterica* serovar Typhimurium str. ATCC 14028S/DSM 19587 (4,964,097 bp)	5 × 10^7^	0.062	Gram −
*Staphylococcus aureus* subsp. *aureus* NCTC 8325 (2,821,361 bp)	5 × 10^8^	0.352	Gram +
**Viruses**			
Bovine alphaherpesvirus 1 (135,098 *)	2.41 × 10^9^	<0.001	ds DNA
Border disease virus isolate Gifhorn (12,325 bp)	6 × 10^6^	<0.001	ssRNA
Bovine viral diarrhea virus type 1 isolate NADL (12,578 bp)	3 × 10^5^	<0.001	ssRNA
**Eukaryota**			
*Cryptosporidium parvum* IOWA II isolate (9,102,324 bp)	1 × 10^6^	0.002	
*Saccharomyces cerevisiae* S288C (12,157,105 bp)	5 × 10^6^	0.015	

**Table 2 microorganisms-08-01861-t002:** Metagenomic datasets and corresponding wet-lab protocols. All protocols are detailed in the Appendix A. Pre-processing protocols were bead beating based protocols with TissueLyser (BBTL), no-processing (NO_PP), extraction with chloroform/butanol, Cat-Floc T, OmniCleave Endonuclease (PEGO), centrifugation (C), liquid nitrogen, centrifugation and filtration in HPF Millex filter (HCFHN), and CryoPrep Covaris (CP). The pre-processing protocols were categorized for the biostatistical analysis as no-processing (NO_PP), bead beating based protocols using TissueLyser (BBTL) and other pre-processing protocols (OTHER_PP). The DNA was extracted by QIAamp Fast DNA Stool and DNeasy Power Soil (categorized as OTHER-EXD), DNeasy Power Food Microbial Kit (PowerFood), QIAamp DNA Mini Kit with or without Sequence-Independent, Single-Primer-Amplification (SISPA), and QIAamp UCP Pathogen Mini Kit (categorized as QIAamp). The RNA was extracted by the RNeasy Mini Kit with or without TRIzol and SISPA (categorized as RNeasy Mini), QIAamp Viral RNA Mini Kit (QIAampV), NucliSENS^®^ miniMAG^®^ and Direct-zol RNA Kit (categorized as OTHER_EXR). The cDNA was generated using the cDNA Synthesis System Kit (cDNA SS) or the SuperScript IV Reverse Transcriptase (SS IV RT). The libraries were prepared by Nextera™ XT DNA Library Prep kit, Nextera™ DNA Flex Library Preparation kit (categorized as NexteraXT), NEBNext^®^ Ultra™ II DNA Library Prep Kit for Illumina^®^, TruSeq^®^ DNA Library Prep Kit and GeneRead DNA Library kit (categorized as OTHER_L). The sequencing platforms were NextSeq500, HiSeq2500 (categorized as NextSeq500), MiniSeq, MiSeq, and Ion Torrent S5XL (categorized as OTHER_SP).

Metagenome Dataset	Nucleic Acid	Pre-Processing	Category Label	Extraction Kit	Category Label	cDNA Generation	Library Kit	Category Label	Sequencing Strategy	Read Length	Sequencing Platform	Category Label	Gbp	Workflow Label
M06	DNA	BBTL	BBTL	QIAamp Fast DNA Stool	OTHER_EXD		NexteraXT	NexteraXT	Paired-end	120	NextSeq 500	NextSeq 500	1.34	WF1
M15	DNA	BBTL	BBTL	QIAamp Fast DNA Stool	OTHER_EXD		NexteraXT	NexteraXT	Paired-end	120	NextSeq 500	NextSeq 500	2.2	WF1
M24	DNA	NO_PP	NO_PP	DNeasy PowerSoil	OTHER_EXD		NexteraXT	NexteraXT	Paired-end	150	NextSeq 500	NextSeq 500	8.95	WF2
M29	DNA	NO_PP	NO_PP	DNeasy PowerSoil	OTHER_EXD		Nextera Flex	NexteraXT	Paired-end	150	MiniSeq	OTHER_SP	3.43	WF2
M38	DNA	NO_PP	NO_PP	DNeasy PowerSoil	OTHER_EXD		NexteraXT	NexteraXT	Paired-end	150	NextSeq 500	NextSeq 500	12.2	WF2
M33	RNA	PEGO	OTHER_PP	NucliSENS MiniMag	OTHER_EXR	SS IV RT	NEBNext	OTHER_L	Paired-end	150	MiSeq	OTHER_SP	0.92	WF3
M34	RNA	PEGO	OTHER_PP	NucliSENS MiniMag	OTHER_EXR	SS IV RT	NEBNext	OTHER_L	Paired-end	150	MiSeq	OTHER_SP	0.86	WF3
M16	DNA	BBTL	BBTL	DNesasy PowerFood	PowerFood		NexteraXT	NexteraXT	Paired-end	120	NextSeq 500	NextSeq 500	1.83	WF4
M18	DNA	BBTL	BBTL	DNesasy PowerFood	PowerFood		NexteraXT	NexteraXT	Paired-end	120	NextSeq 500	NextSeq 500	2.50	WF4
M23	RNA	BBTL	BBTL	Direct-zol RNA	OTHER_EXR	cDNA SS	TruSeq	OTHER_L	Paired-end	150	NextSeq 500	NextSeq 500	11.01	WF5
M30	DNA	C	OTHER_PP	DNesasy PowerFood	PowerFood		NexteraXT	NexteraXT	Paired-end	150	NextSeq 500	NextSeq 500	7.97	WF6
M31	DNA	C	OTHER_PP	DNesasy PowerFood	PowerFood		NexteraXT	NexteraXT	Paired-end	150	NextSeq 500	NextSeq 500	9.81	WF6
M32	DNA	C	OTHER_PP	DNesasy PowerFood	PowerFood		NexteraXT	NexteraXT	Paired-end	150	NextSeq 500	NextSeq 500	8.44	WF6
M07	DNA	NO_PP	NO_PP	QIAamp UCP Pathogen	QIAamp		NexteraXT	NexteraXT	Paired-end	120	NextSeq 500	NextSeq 500	1.03	WF7
M10	DNA	NO_PP	NO_PP	QIAamp UCP Pathogen	QIAamp		NexteraXT	NexteraXT	Paired-end	120	NextSeq 500	NextSeq 500	1.77	WF7
M26	DNA	BBTL	BBTL	QIAamp	QIAamp		TruSeq	OTHER_L	Paired-end	150	NextSeq 500	NextSeq 500	12.2	WF8
M36	DNA	BBTL	BBTL	QIAamp	QIAamp		TruSeq	OTHER_L	Paired-end	250	HiSeq 2500	NextSeq 500	8.88	WF8
M25	RNA	NO_PP	NO_PP	QIAamp Viral RNA	QIAamp	cDNA SS	NexteraXT	NexteraXT	Paired-end	150	NextSeq 500	NextSeq 500	8.64	WF9
M37	RNA	NO_PP	NO_PP	QIAamp Viral RNA	QIAamp	SS IV RT	NexteraXT	NexteraXT	Paired-end	200	MiSeq	OTHER_SP	8.84	WF10
M08	DNA	HCFHN	OTHER_PP	QIAamp + SISPA	QIAamp		NexteraXT	NexteraXT	Paired-end	300	MiSeq	OTHER_SP	1.99	WF11
M11	DNA	HCFHN	OTHER_PP	QIAamp + SISPA	QIAamp		NexteraXT	NexteraXT	Paired-end	300	MiSeq	OTHER_SP	2.07	WF11
M13	DNA	HCFHN	OTHER_PP	QIAamp + SISPA	QIAamp		NexteraXT	NexteraXT	Paired-end	300	MiSeq	OTHER_SP	2.62	WF11
M27	DNA	CP	OTHER_PP	QIAamp	QIAamp		GeneRead	OTHER_L	Single-end	250	Ion Torrent S5XL	OTHER_SP	2.14	WF12
M19	RNA	BBTL	BBTL	RNeasy Mini kit	RNeasy Mini	cDNA SS	NexteraXT	NexteraXT	Paired-end	120	NextSeq 500	NextSeq 500	4.95	WF13
M20	RNA	BBTL	BBTL	RNeasy Mini kit	RNeasy Mini	cDNA SS	NexteraXT	NexteraXT	Paired-end	120	NextSeq 500	NextSeq 500	5.06	WF13
M12	RNA	HCFHN	OTHER_PP	RNeasy Mini + SISPA	RNeasy Mini	SS IV RT	NexteraXT	NexteraXT	Paired-end	300	MiSeq	OTHER_SP	2.63	WF14
M28	RNA	CP	OTHER_PP	RNeasy Mini kit	RNeasy Mini	cDNA SS	GeneRead	OTHER_L	Single-end	250	Ion Torrent S5XL	OTHER_SP	2.25	WF15

**Table 3 microorganisms-08-01861-t003:** Number of reads in each metagenomic dataset obtained from spiked salmon. For each dataset, the percentage of reads assigned to Eukaryota, Bacteria, Viruses, Archaea, the number of reads belonging to the microorganisms of the mock community, and their percentage in relation to the total number of reads are detailed.

Metagenomic Dataset Label	N. Reads	% Eukaryota	% Bacteria	% Viruses	% Archaea	N. Reads Mock Community (%)
M06	1,644,354	14.631	85.004	0.319	0.025	875,934 (53.27)
M07	847,827	20.857	78.846	0.249	0.036	369,389 (43.57)
M08	742,029	80.338	19.050	0.325	0.286	20,440 (2.76)
M10	1,071,120	25.539	74.097	0.326	0.028	388,946 (36.31)
M11	730,610	80.567	18.823	0.309	0.299	3763 (0.52)
M12	537,347	79.502	19.747	0.469	0.264	11,073 (2.06)
M13	993,696	72.565	26.222	0.488	0.725	10,600 (1.07)
M15	1,508,927	21.857	77.846	0.254	0.033	635,158 (42.09)
M16	1,206,052	24.369	75.370	0.216	0.036	496,223 (41.14)
M18	1,867,262	19.308	80.394	0.259	0.027	681,708 (35.61)
M19	16,187	31.951	67.770	0.031	0.043	826 (5.1)
M20	16,702	64.459	35.475	0.048	0.018	1225 (7.33)
M23	82,614	45.051	54.713	0.171	0.036	21,891 (26.50)
M24	3,304,160	31.519	68.311	0.117	0.046	1,373,626 (41.57)
M25	1,173,758	45.668	54.015	0.222	0.073	353,797 (30.14)
M26	4,803,071	40.370	59.363	0.230	0.028	1,332,603 (27.75)
M27	911,713	46.621	53.249	0.104	0.021	205,421 (22.53)
M28	2340	72.393	27.564	0.043	0	203 (8.68)
M29	1,507,815	35.777	64.055	0.115	0.045	430,397 (28.54)
M30	3,680,106	14.271	85.576	0.122	0.022	2,217,739 (60.26)
M31	3,360,140	17.916	81.949	0.104	0.021	1,883,366 (56.05)
M32	4,884,497	11.043	88.809	0.119	0.020	2,769,020 (56.69)
M33	203,116	15.636	45.945	38.344	0.072	2156 (1.06)
M34	267,871	2.957	93.85	3.017	0.084	2231 (0.83)
M37	1,735,966	48.139	51.513	0.265	0.076	511,159 (29.45)
M36	17,120,850	4.586	95.202	0.148	0.057	9,405,164 (54.93)
M38	5,803,430	21.492	78.322	0.135	0.044	2,329,995 (40.15)

**Table 4 microorganisms-08-01861-t004:** Ranking of the DNA metagenomic datasets obtained testing spiked salmon in relation to their similarity to the expected composition of the mock community. The distance from the expected mock community composition is measured as Bray–Curtis dissimilarity.

Metagenomic Dataset	Distance from the Mock Community	Rank
M36	0.207	1
M38	0.337	2
M24	0.345	3
M30	0.361	4
M32	0.370	5
M31	0.395	6
M07	0.435	7
M16	0.447	8
M27	0.464	9
M26	0.488	10
M18	0.495	11
M10	0.496	12
M15	0.498	13
M29	0.544	14
M08	0.556	15
M06	0.610	16
M11	0.646	17
M13	0.790	18

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
