# Peer review of "Metagenomics-Based Proficiency Test of Smoked Salmon Spiked with a Mock Community"

_microorganisms, 2020, doi:10.3390/microorganisms8121861_

Round 1

Reviewer 1 Report

In this manuscript, the authors launched a proficiency test project to examine the detection of shotgun metagenomic sequencing on food-borne pathogens. They made a very complete examination of different settings including 27 combinations of sample preprocessing, extraction kit, cDNA synthesis, library preparation kit etc. and test their effects on the detection of Eukaryotes, Bacteria and Virus. As a result, they found the shotgun metagenomic sequencing is able to detect and quantify the pathogens, and wet-lab process has huge effects on the sensitivity and quantification of the detection. For example, the pre-processing protocol significantly affected the detected abundances of C. parvum and E. coli, 314 etc. in the mock community.

The results are interesting and have practical significance guiding the food-borne pathogen detection. My major concern is that the accuracy of taxonomic classification based on single reads is as low as family level and insufficient to assign reads to the species of the mock community. This is the defect of shotgun metagenomic detection in comparison to marker gene amplicon sequencing (e.g. 16S rRNA). The advantage of shotgun metagenomic detection over amplicon sequencing to detect virus and detect multiple domains simultaneously is, however, not mentioned.

Detailed comments:

Line 100. An aliquot of cold smoked salmon was cut in small pieces (approximately 1-2 mm in 101 width/length/depth) using a sterile scalpel and sterile petri dish.

The contamination microorganism DNA that were originally in the salmon tissue before spiking the mock community may affect the quantification of the mock community species. Because of the insufficient resolution of read-based taxonomic classification, close relatives, for example, other species of the same genus, would be misclassified as the species of the mock community. These should be discussed in the text.

Table 1.

Saccharomyces cerevisiae is usually diploid (https://academic.oup.com/femsyr/article/10/6/757/539554) with two sets of genome. Its expected relative abundance should be 13 times as much as the Cryptosporidium parvum which is haploid rather than 7.5 times. Can you please recheck it?

Line 167. In 9 samples the target nucleic acid was RNA (Table 2).

Are these RNA extraction protocols designed to remove genomic DNA?

Line 265. those to viruses between 0.005 % (M21) to 38.344 % (M33)

Can you discuss why is virus in some RNA-targeted samples e.g. (M33) so abundant while much less in other RNA-targeted samples (e.g. M19)?

Figure 1. Why were RNA virus (Border disease virus and Bovine viral diarrhea virus) detected in DNA-targeted samples? RNA could be extracted using DNA extraction methods, but it may not link to sequencing index and adaptors in the library preparation.

Were Typhimurium and Freudenreichii not detected?

MG-RAST can annotate metagenomic data using either read-based or assembly-based methods. Are these taxonomic results based on read or contig? If based on read, the resolution of single-read search against database are quite low (family level) and cannot reach species level due to the short length. Therefore, the numbers of reads assigned to the mock community at species level are actually a rough estimation. These should be discussed in the text.

Author Response

Reviewer 1

In this manuscript, the authors launched a proficiency test project to examine the detection of shotgun metagenomic sequencing on food-borne pathogens. They made a very complete examination of different settings including 27 combinations of sample preprocessing, extraction kit, cDNA synthesis, library preparation kit etc. and test their effects on the detection of Eukaryotes, Bacteria and Virus. As a result, they found the shotgun metagenomic sequencing is able to detect and quantify the pathogens, and wet-lab process has huge effects on the sensitivity and quantification of the detection. For example, the pre-processing protocol significantly affected the detected abundances of C. parvum and E. coli, 314 etc. in the mock community.

Question 1: The results are interesting and have practical significance guiding the food-borne pathogen detection. My major concern is that the accuracy of taxonomic classification based on single reads is as low as family level and insufficient to assign reads to the species of the mock community. This is the defect of shotgun metagenomic detection in comparison to marker gene amplicon sequencing (e.g. 16S rRNA).

Answer 1: The authors are not sure why the Reviewer refers to the family level while the species level was reached. Please note that in a recent paper (Brumfield, Kyle D., et al. "Microbial resolution of whole genome shotgun and 16S amplicon metagenomic sequencing using publicly available NEON data." Plos one 15.2 (2020): e0228899) available at the following link https://journals.plos.org/plosone/article/file?id=10.1371/journal.pone.0228899&type=printable, whole genome shotgun metagenomic sequencing and 16S amplicon sequencing were compared for application to environmental metagenomics. The paper shows that metagenomics was able to detect and identify more genera of bacteria, archaea, viruses, and eukaryota compared to 16S amplicon sequencing although other analytical software, in addition to MG-RAST, may be required to resolve taxonomic decisions below genus, i.e., species, strain, and sub strain. It says “may be required” because MG RAST provides rarefaction curves as the total number of distinct species annotations, a function of the number of sequencing reads. In our study the results at species level were considered feasible because the spiked species were known but see also answer 3 below addressing your concern.

Question 2: The advantage of shotgun metagenomic detection over amplicon sequencing to detect virus and detect multiple domains simultaneously is, however, not mentioned.

Answer 2: In the abstract, lines 40-42 state “…proving the suitability of shotgun metagenomic sequencing as genomic tool to detect microorganisms belonging to different domains in the same food matrix”. Moreover, in the discussion lines 372-374 state “This PT was organized with the aim to compare the suitability of different metagenomic wet-lab protocols to both detect and quantify the relative abundance of microorganisms belonging to different domains experimentally spiked in cold smoked salmon.” However, a further highlight has been added in the conclusion in lines 484-485 to address the Reviewer comment.

Detailed comments:

Question 3: Line 100. An aliquot of cold smoked salmon was cut in small pieces (approximately 1-2 mm in 101 width/length/depth) using a sterile scalpel and sterile petri dish.

The contamination microorganism DNA that were originally in the salmon tissue before spiking the mock community may affect the quantification of the mock community species. Because of the insufficient resolution of read-based taxonomic classification, close relatives, for example, other species of the same genus, would be misclassified as the species of the mock community. These should be discussed in the text.

Answer 3: Although it has been shown that MG-RAST has good sensitivity and precision when assigning reads to the species of a mock community (Peabody et al 2015) we acknowledge that MG-RAST itself suggests against the usage of the species taxonomic level and that the bioinformatic processing of the data may have affected the quantification of the species of the mock community, especially in the presence of possible contamination microorganisms that are close relatives of the mock community species. As requested by the Reviewer this is now discussed in the text in lines 431-436 and the reference Peabody at al. 2015 has been added with number 41.  

Question 4 Table 1.

Saccharomyces cerevisiae is usually diploid (https://academic.oup.com/femsyr/article/10/6/757/539554) with two sets of genome. Its expected relative abundance should be 13 times as much as the Cryptosporidium parvum which is haploid rather than 7.5 times. Can you please recheck it?

Answer 4: The reviewer is 100% right. The data has been checked and indeed it was wrong. The analysis has been run again and as a result the followings changes have been added: (1) the distance values in Table 4 and TableS4 have been updated. They are slightly different to the previous ones and the ranking in both Table 4 and Table S4 did not change; (2) Fig 1 and 2 have been updated. Thanks to the Reviewer for the deep look into this.  

Question 5: Line 167. In 9 samples the target nucleic acid was RNA (Table 2).

Are these RNA extraction protocols designed to remove genomic DNA?

Answer 5: As the reviewer can easily imagine the paper does not comment on each single kit used by the participants to avoid giving the impression to endorse specific commercial kits. Let say that there is no RNA isolation method that consistently produces RNA free from genomic DNA without the use of DNase. Among the kits applied, the RNeasy Kit provides high-quality RNA with minimum copurification of DNA, while Direct-zol RNa MiniPreps for instance, might result in a purer extracted RNA. This kind of comparisons must be addressed in target studies in which the single variable investigated is the extraction kit.

Question 6: Line 265. those to viruses between 0.005 % (M21) to 38.344 % (M33)

Can you discuss why is virus in some RNA-targeted samples e.g. (M33) so abundant while much less in other RNA-targeted samples (e.g. M19)?

Answer 6: Besides typing errors which have been corrected, the RNA-targeted samples in which viruses were hard to detect where those obtained with WF13 and WF15 in which RNA was extracted using the RNeasy mini kit and this has been highlighted now in the revision in lines 267-269. However, the authors are not able to conclude that the extraction kit is the only reason for that.

Question 7: Figure 1. Why were RNA virus (Border disease virus and Bovine viral diarrhea virus) detected in DNA-targeted samples? RNA could be extracted using DNA extraction methods, but it may not link to sequencing index and adaptors in the library preparation.

Answer 7: As shown in Table S2, RNA virus (Border disease virus and Bovine viral diarrhea virus) were not detected in the DNA-targeted samples. They were in the legend as part of the mock community but have been deleted because the authors agree with the Reviewer they should not be there. Thanks for the advice.

Question 8: Were Typhimurium and Freudenreichii not detected?

Answer 8: In the Figure S Typhimurium is labelled as Salmonella enterica because that was the only serotype of Salmonella spiked in the mock community and it was detected as highlighted by the blue; P. freudenreichii was also detected as highlighted by the green color.   

Question 9: MG-RAST can annotate metagenomic data using either read-based or assembly-based methods. Are these taxonomic results based on read or contig? If based on read, the resolution of single-read search against database are quite low (family level) and cannot reach species level due to the short length. Therefore, the numbers of reads assigned to the mock community at species level are actually a rough estimation. These should be discussed in the text.

Answer 9: Please see answer 3 above on discussion added to answer the Reviewer comment in lines 431-436

Reviewer 2 Report

Shotgun metagenomic sequencing of cold smoked salmon experimentally spiked with six bacteria(1 parasite, 1 yeast, 1 DNA virus ,2 RNA viruses)was evaluated in the present study. Participants entered in the study  applied their in-house wet-lab workflows to obtain the metagenomic datasets. A total of 27 datasets were analyzed. Analysis were done by  using MG-RAST. Among the variables investigated in the workflows those impacting on the abundance of one or  more microorganisms spiked in the salmon were the pre-processing protocol, the DNA extraction protocol, the library preparation strategy and the sequencing platform. 

It is a global study coming from different European Labs and results highlight the fact that differences in bacterial ,parasites and viruses abundances were observed in the metagenomic datasets. The study aims to compare the suitability of different metagenomic wet-lab
protocols to both detect and quantify the relative abundance of microorganisms belonging to
different domains as Shotgun metagenomics could be used in  food-borne outbreaks detection and risk assessment of food-borne pathogens.

However, through the study it seems that there is a need of uniformization of the different  compare the suitability of different metagenomic wet-lab protocols to both detect and quantify the relative abundance of microorganisms.Through the study it seems that there is a need of uniformization of the pre-processing protocol, the DNA extraction protocol, the library preparation strategy and the sequencing platform. 

It is a well written paper with a heavy experimental part and relevant statitistics .

My Suggestion is to ACCEPT the paper and publish it in its present form as it should be interesting for evaluating protocols not only for salmon studies but also other foods stuff.

Author Response

My Suggestion is to ACCEPT the paper and publish it in its present form as it should be interesting for evaluating protocols not only for salmon studies but also other foods stuff.

Answer: thanks for the appreciation of our work.